# Topology-transformable block copolymers based on a rotaxane structure: change in bulk properties with same composition

Hiroki Sato [1], Daisuke Aoki [1], Hironori Marubayashi [1], Satoshi Uchida[1], Hiromitsu Sogawa [1], Shuichi Nojima[1], Xiaobin Liang [1], Ken Nakajima [1], Teruaki Hayakawa [2] & Toshikazu Takata [1,3,4 ✉]

The topology of polymers affects their characteristic features, i.e., their microscopic structure and macroscopic properties. However, the topology of a polymer is usually fixed during the construction of the polymer chain and cannot be transformed after its determination during the synthesis. In this study, topology-transformable block copolymers that are connected via rotaxane linkages are introduced. We will present systems in which the topology transformation of block copolymers changes their 1) microphase-separated structures and 2) macroscopic mechanical properties. The combination of a rotaxane structure at the junction point and block copolymers that spontaneously form microphase-separated structures in the bulk provides access to systems that cannot be attained using conventional covalent bonds.

[1] Department of Chemical Science and Engineering, Tokyo Institute of Technology, Ookayama, Meguro, Tokyo 152-8552, Japan. [2] Department of Materials Science and Engineering, Tokyo Institute of Technology, Ookayama, Meguro, Tokyo 152-8552, Japan. [3] JST-CREST, Ookayama, Meguro, Tokyo 152-8552, Japan. [4] Graduate School of Advanced Science and Engineering, Hiroshima University, Kagamiyama, Higashi-Hiroshima, Hiroshima 739-8527, Japan. ✉email: takatats@hiroshima-u.ac.jp

Topology is essential in chemistry to describe the features of not only single molecules, but also molecular assemblies, i.e., macroscopic-level features; thus, topological transformations have attracted great attention for their potential applications. For example, dynamic conformational changes, such as the light-induced *trans–cis* photoisomerization of azobenzene, can lead to changes in optical properties and macroscopic contraction or bending in response to external stimuli[1,2]. Irie et al. have demonstrated the deformation of molecular crystals of diarylethene due to its topological transition from the open-ring to the closed-ring isomer upon photoirradiation[3]. However, such transformable molecules are still limited mainly to small molecules. Topology also dominates the characteristic features of polymers, especially in multicomponent systems such as block copolymers, and various multicomponent polymer topologies, i.e., linear, cyclic, branched, and grafted have been synthesized and characterized[4–7]. For example, Matsushita et al. have reported that triblock copolymers form different microphase-separated structures based on their topology[8,9]. Differences in the microphase-separated structure of block copolymers, are well-known to affect their macroscopic mechanical properties[10–12]. For example, ABA triblock copolymers work as thermoplastic elastomers (TPEs) which are widely used in various fields, but this functionality is not attained by AB diblock copolymers. Therefore, polymers whose properties can be changed by inducing a transformation in their topology represent an attractive class of polymer systems. As the variety of possible polymer chains is virtually infinite, the concept of topology transformation in polymers would enable the preparation of a wide variety of smart materials whose properties could be modified via topological transitions. However, the synthesis of polymers that can undergo a topological change without a change in their composition remains challenging. Recently, we have reported various topology-transformable polymers with linear-to-star[13,14] and linear-to-cyclic[15–18] transitions via the introduction of a rotaxane linkage at the junction point of the polymer chain.

In this work, the initial microphase-separated structure of the multicomponent polymer system is converted by a polymer topology transformation via a movable rotaxane linkage at the junction point. As the microphase-separated structure of a polymer has a critical effect on its macroscopic properties, two types of topology-transformable block copolymers with carefully designed compositions are prepared and characterized (Fig. 1): (1) Rotaxane-linked star terpolymers ABC (Fig. 1a), which are designed to undergo a transition in their microphase-separated structure and visually demonstrate the concept of topology-transformable polymers in the bulk state, and (2) $A_2B_2$ tetra-block copolymers (Fig. 1b), which are designed to undergo a change in their mechanical properties upon topology transformation, thus demonstrating an application of topology-transformable block copolymers.

## Results

**Topology transformation of a star ABC terpolymer to a linear ABC terpolymer.** An ABC terpolymer consisting of poly(valerolactone) (PVL), polystyrene (PS), and polydimethylsiloxane (PDMS) was designed in order to confirm potential changes in the microphase-separated structure. As shown in Fig. 2, the

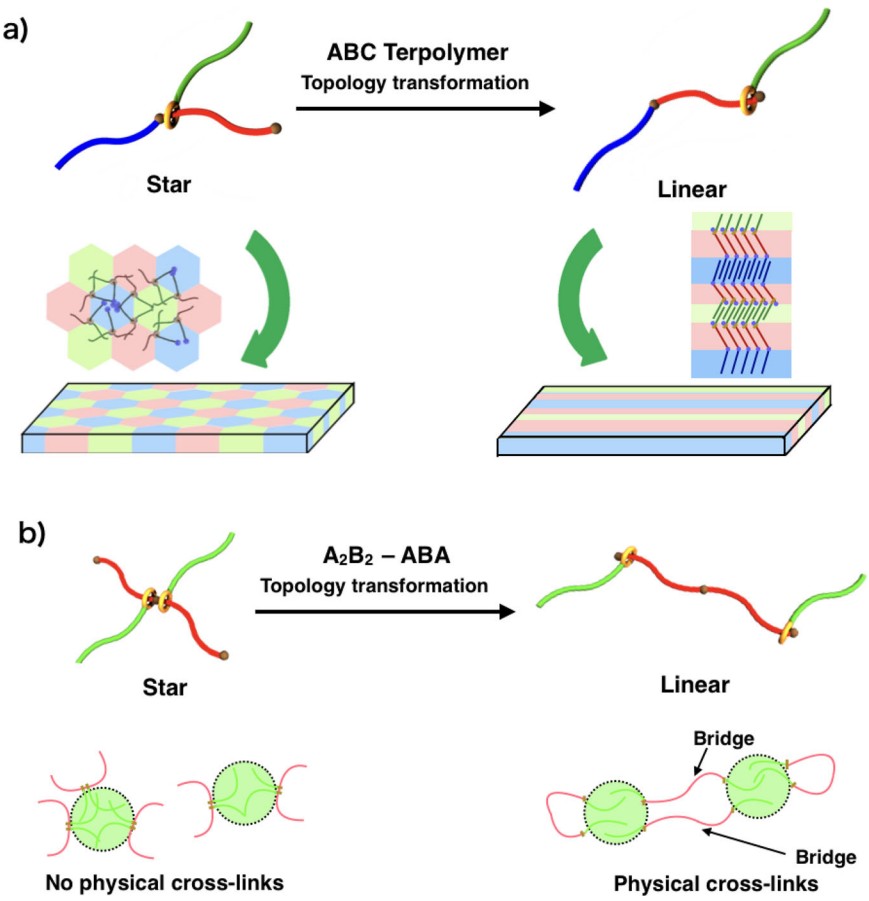

**Fig. 1 Two types of topology-transformable block copolymers. a** A rotaxane-linked star ABC terpolymer designed to undergo a change in its microphase-separated structure and visually demonstrate the concept of topology-transformable polymers. **b** An $A_2B_2$ tetra-block copolymer designed to demonstrate a change in its mechanical properties via topology transformation.

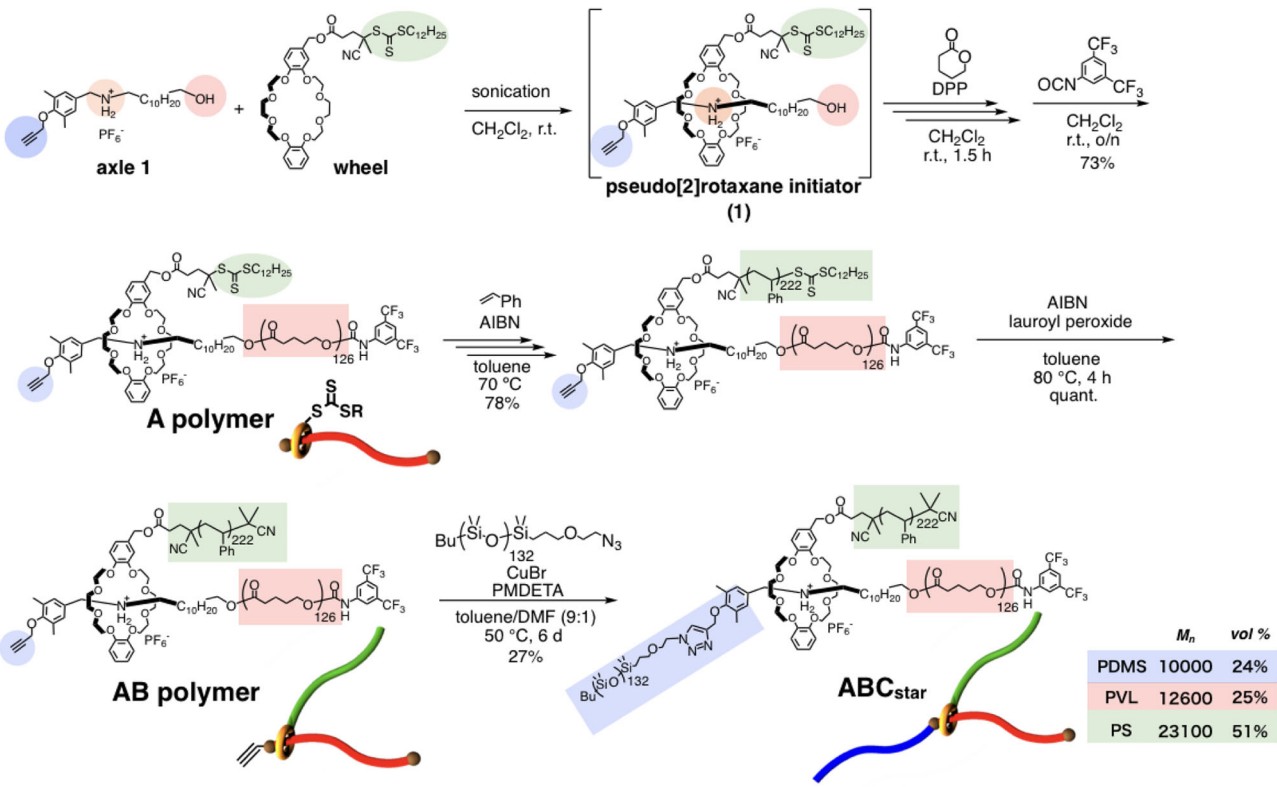

**Fig. 2 Synthetic scheme of topology-transformable ABC star terpolymer.** The ABC star terpolymer (**ABC_star**) consisting of poly(valerolactone) (PVL), polystyrene (PS), and polydimethylsiloxane (PDMS).

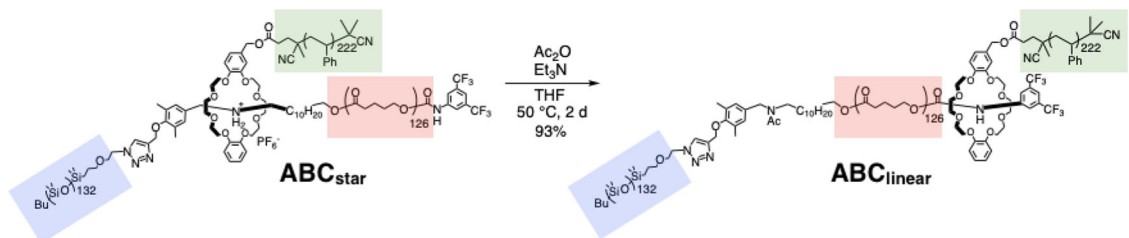

**Fig. 3 Polymer topology transformation from star to linear.** Transformation of the ABC terpolymer from a star-shaped (**ABC_linear**) to a linear (**ABC_linear**) topology.

polymer was synthesized according to our previously developed method[14]; (1) Living ring-opening polymerization of δ-valerolactone (δ-VL) using a pseudo-[2]rotaxane initiator (**1**), (2) reversible addition−fragmentation chain transfer (RAFT) polymerization of styrene onto the crown ether, and (3) copper-catalyzed alkyne−azide cycloaddition between the ethynyl group at the end of the axle component and azide-terminated PDMS. The resulting ABC terpolymer was fully characterized using $^1$H NMR spectroscopy and GPC, which demonstrated the successful synthesis of the ABC terpolymer with a 12600(PVL): 23100(PS):10000(PDMS) molecular mass ratio (estimated based on $^1$H NMR data) and a volume fraction ratio of 25(PVL): 51(PS):24(PDMS). The yield of copper-catalyzed alkyne−azide cycloaddition was relatively low. Steric hindrance around alkyne group might result in less reactivity. The purification process using preparative GPC to obtain the perfectly pure ABC terpolymer also affects the decrease in yield. As shown in Fig. 3, the transformation of the ABC terpolymer from a star-shaped (**ABC_star**) to a linear (**ABC_linear**) topology, which is due to the interaction change from *sec*-ammonium/crown ether to urethane linkage/crown ether[19–21], was achieved via the acetylation of the

*sec*-ammonium moiety in the center of the polymer chain in **ABC_star**. This reaction proceeded quantitatively.

The increase in hydrodynamic volume after acetylation supports the transformation from a star to a linear topology (Supplementary Fig. 5). DSC measurements before and after the topology transformation revealed that the melting enthalpy, a measure of crystallinity, of both crystalline blocks, i.e., PDMS and PVL blocks, increased after the topology transformation (Supplementary Fig. 7). The morphologies of the ABC terpolymers before and after the topology transformation were characterized using scanning transmission electron microscopy (STEM) and small-angle X-ray scattering (SAXS). Fig. 4 shows the bright-field STEM images of each polymer, which clearly demonstrate their different microphase-separated structures. The STEM image of **ABC_star** shows a spherical structure in which the distance between neighboring spheres is ca. 24 nm, and that of **ABC_linear** depicts a mixed structure of lamellar ($d \approx 37$ nm) and cylinder-like structures (only lamellar structure in Fig. 4; Supplementary Fig. 9 shows the image in a wide field of view). The dark areas in both images were assigned to the PDMS microdomains having relatively electron-rich atoms (silicon atoms); this assignment was

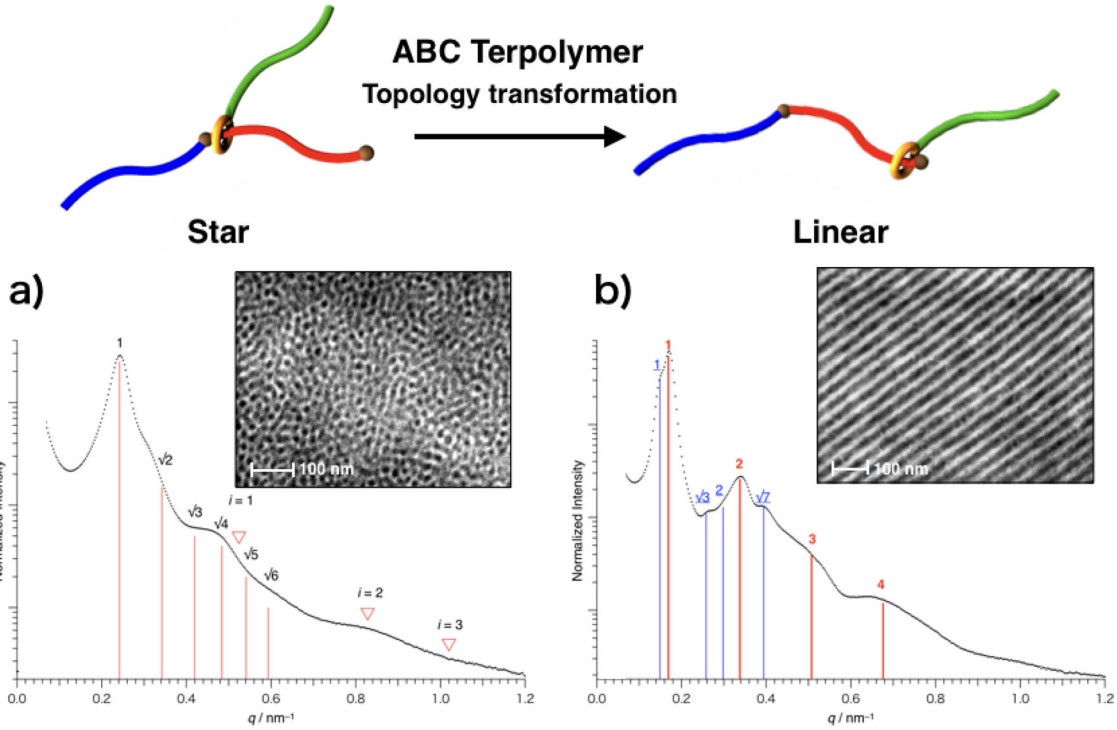

**Fig. 4 SAXS profiles and bright-field STEM images before (ABC_star) and after (ABC_linear) the topology transformation. a ABC_star** and (**b**) **ABC_linear.** In the SAXS profile of **ABC_linear**, the peak positions of lamellar and cylinder-like structures are indicated by red and blue, respectively.

also supported by the energy dispersive X-ray spectroscopy (EDS) map in the same viewing field as STEM (Supplementary Fig. 10). After the topology transformation, the spherical PDMS microdomains formed by **ABC_star** were translated into a mixed structure of lamellar and cylinder-like structures in **ABC_linear**. The SAXS profiles before and after the topology transformation are also shown in Fig. 4. The SAXS measurements were performed at room temperature (the same as STEM), 70 °C [> melting temperature ($T_m$) of PVL block (Supplementary Fig. 7)], and 120 °C [> glass-transition temperature ($T_g$) of PS block (Supplementary Fig. 7)]. The profiles at 70 °C are shown as representative, because the peak intensities increased (especially in **ABC_linear**) while the primary peak positions were almost the same with increasing temperature to 70 °C (Supplementary Figs. 11 and 12). The spacings calculated from the primary peak positions (26 nm for **ABC_star**; 37 nm for lamellar structure in **ABC_linear**) are in good agreement with those obtained from STEM images (24 nm for **ABC_star**; 37 nm for lamellar structure in **ABC_linear**).

Here, the temperature-dependent changes in ordered morphologies of two ABC terpolymers are discussed based on simultaneous SAXS and wide-angle X-ray diffraction (WAXD, Supplementary Fig. 17) data. It should be noted that the SAXS profile of **ABC_star** is clearly different from that of **ABC_linear** irrespective of temperature (Supplementary Figs. 11–16). In the case of **ABC_star**, SAXS profiles show only slight change with increasing temperature from room temperature to 70 °C (Supplementary Fig. 11). In contrast, SAXS profiles of **ABC_linear** show a drastic change with increasing temperature to 70 °C (Supplementary Fig. 12). This clear difference between **ABC_star** and **ABC_linear** would be attributed to a large difference in the crystallinity of PVL block ($\chi_{c,PVL}$), which was evaluated from WAXD profiles. In the case of **ABC_star**, only a small amount of PVL block can crystallize ($\chi_{c,PVL} = 14\%$), so that the microdomain structure should be less affected by the crystallization of PVL block. On the contrary, highly crystalline PVL blocks in

**ABC_linear** ($\chi_{c,PVL} = 43\%$) should deform the microdomain structure (mainly soft PDMS domain)[22]. The drastic increase in SAXS peak intensity of **ABC_linear** indicates the ordering of distorted microdomain structure by melting of PVL crystals. Because the PS domain is still in the glassy state at 70 °C [< $T_g$ of PS block (95 °C)], structural defects cannot be sufficiently removed. With further increasing temperature to 120 °C (> $T_g$ of PS block), the system of **ABC_linear** would become closer to the equilibrium ordered morphology, which is supported by sharpened and newly-appeared higher-order SAXS peaks (Supplementary Figs. 12 and 16). In addition, the peak shift to lower $q$ (increase in domain spacings) was observed at 120 °C in both **ABC** terpolymers, which can be explained by the ordering of microphase-separated structure (i.e., block chains are more stretched) and thermal expansion of rubbery PS domains.

The microphase-separated structure of **ABC_linear** can be interpreted by two models; (i) Because the interaction between urethane linkage/crown ether in **ABC_linear** is not so strong as *sec*-ammonium/crown ether interaction in **ABC_star**. Therefore, the wheel component might move as temperature increases, resulting in the change in topology depending on the position of wheel component. In most cases, the wheel component should be loosely fixed at the end of PVL chain, but probably some amount of wheel is not located at the end of PVL chain, which seems like a mixture of copolymer topologies. The polymers with similar but different topologies might separately form different morphologies, leading to a mixture of morphologies. (ii) Another structural model is a single ordered morphology, a hexagonally perforated layer (HPL) morphology, in which lamellar and cylinder structures can be observed depending on viewing directions[23]. Therefore, if some grains with different orientations are adjacent to each other, both lamellar and cylinder structures are observed at the same time. However, STEM images are not clear enough to identify the morphology as HPL, probably because the microdomain structure was distorted by the crystallization of PVL block, as mentioned above. Although the detailed structural

model of **ABC_linear** in the melt remains an open question, we emphasize again that the star-to-linear topology transformation changes the microphase-separated structure.

The concept of topology transformation has previously been discussed and demonstrated in terms of changes in hydrodynamic volume and viscosity as well as NMR data, i.e., topology transformation has been supported only in the solution state[19–21]. In summary, in this study, we successfully demonstrated topology transformation through rotaxane linkages in the bulk state, with different morphologies being observed in the solid state complementarily using SAXS and STEM.

**Topology transformation of an A₂B₂ polymer to an ABA polymer.** In the previous part, we have demonstrated a change in the microphase-separated structure of a polymer upon a star-to-linear topology transformation without any change in the polymer composition. In this part, we focus on the changes in the mechanical properties of a block copolymer in the bulk based on a transformation of its topology. For that purpose, we designed A₂B₂ block copolymers connected by a rotaxane structures that can change from a star-shaped A₂B₂ to a linear ABA topology. The characteristics of TPEs, such as the representative TPE polystyrene–*block*-polybutadiene–*block*-polystyrene (SBS), which has the same elasticity as rubber at normal temperature and can be melted to flow at high temperature like plastics, are derived from the cross-linking effect of a microphase-separated structure

consisting of hard polystyrene segments and soft polybutadiene segments, i.e., physical cross-links anchor the rubbery chains at the service temperature[24–27]. The composition of TPEs inspired us to design a novel system in which a change in the polymer topology dominates the mechanical properties (Fig. 1b).

To incorporate movable rotaxane junction points into a TPE system, we synthesized an A₂B₂ block copolymer consisting of poly(β-methyl-δ-valerolactone) (PmVL) as the soft segment and PS as the hard segment (Fig. 5). The pseudo-[3]rotaxane (**2**), which consists of two *sec*-ammonium moieties as the axle component and two crown ethers with RAFT agents as the wheel components, was used as the initiator for the diphenyl phosphate (DPP)-catalyzed living ring-opening polymerization of β-methyl-δ-valerolactone (MeVL). After the polymerization, 3,5-bis(trifluoromethyl)phenylisocyanate was added to cap the propagating end with a bulky stopper and introduce a urethane moiety at the ω-end on the axle, which acts as a second recognition site at the end of the polymer chain. The obtained linear polymer (**Linear polymer**) was subjected to a RAFT polymerization of styrene onto the crown ether, followed by the removal of the RAFT agent from the polymer terminals using a combination of AIBN and lauroyl peroxide. Characterization using ¹H NMR spectroscopy and GPC (Supplementary Figs. 18 and 19) supported the successful synthesis of an A₂B₂ block copolymer with a 18600(PmVL):17800(PS; total $M_n$ for both PS chains) molecular mass ratio (based on ¹H NMR data). The topology

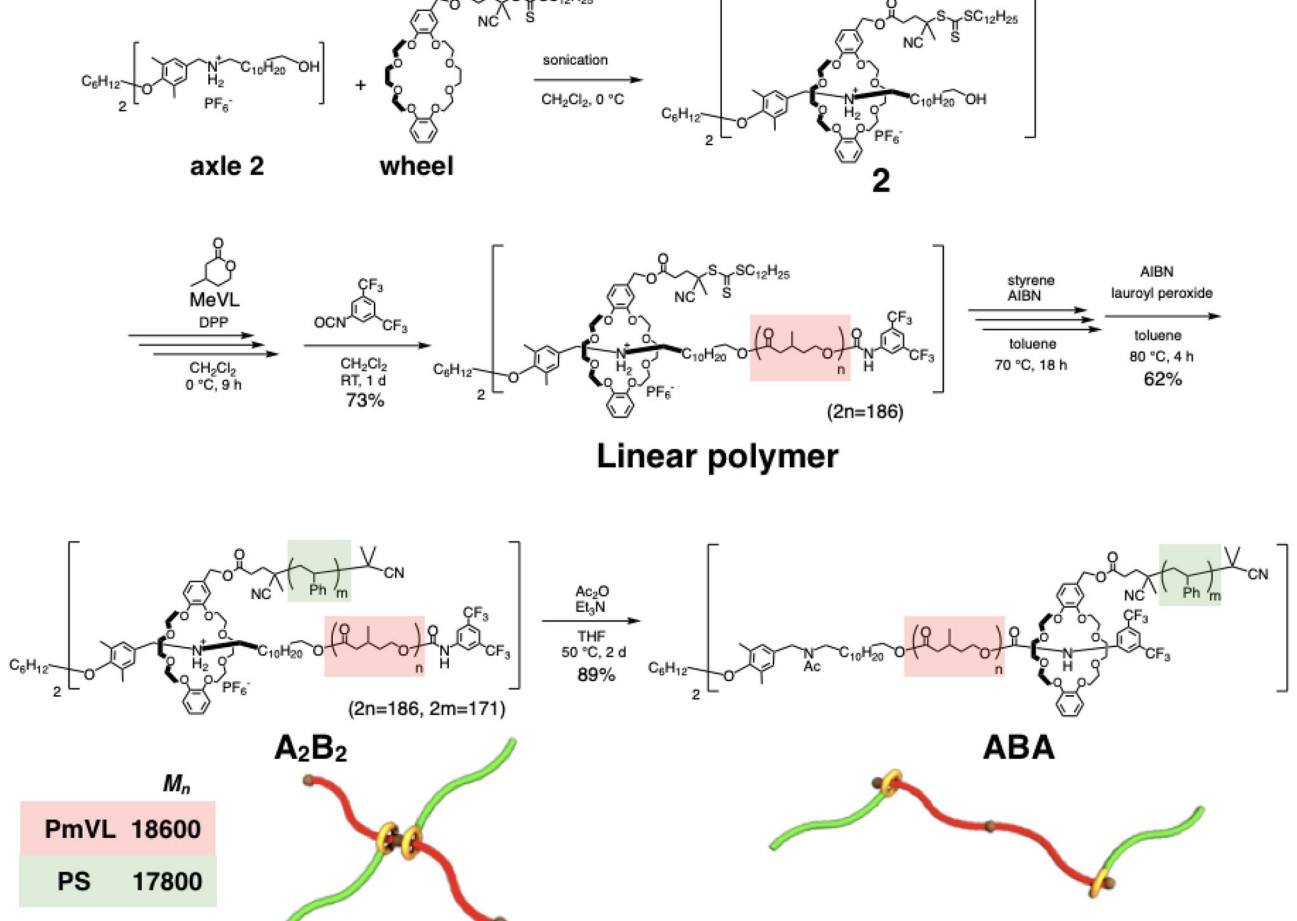

**Fig. 5 Synthetic scheme of a rotaxane-linked A₂B₂ polymer and its transformation into an ABA topology.** Living ring-opening polymerization of MeVL using a pseudo-[3]rotaxane initiator (**2**), reversible addition−fragmentation chain transfer (RAFT) polymerization of styrene onto the crown ether, and topology transformation via acetylation of the *sec*-ammonium moiety at the center of the polymer chain.

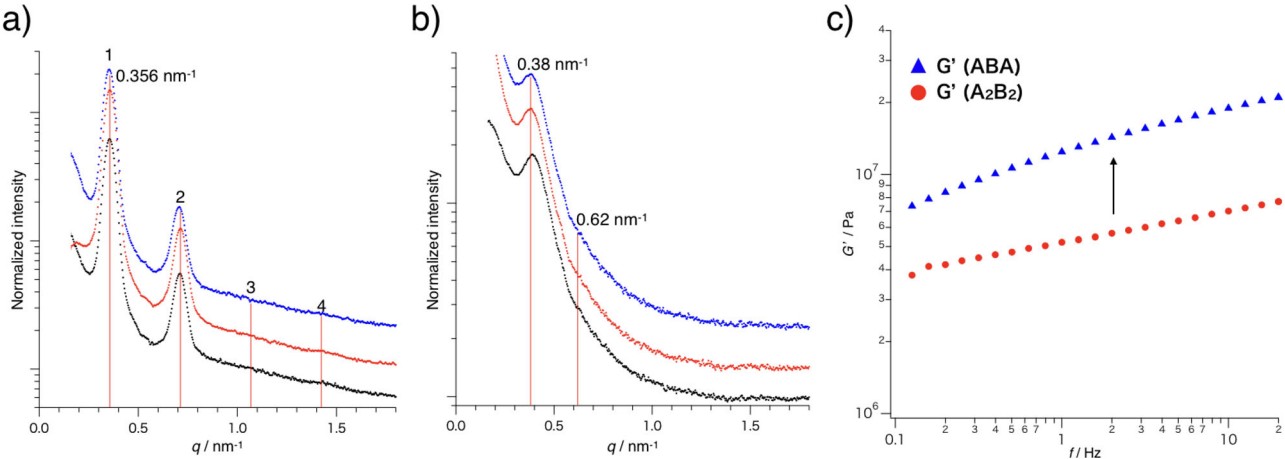

**Fig. 6 SAXS profiles and shear rheology profiles of before and after topology transformation.** SAXS profiles (**a**) before (**A₂B₂**) and (**b**) after (**ABA**) the topology transformation (annealing conditions: black—no annealing; red—100 °C, 12 h; blue—120 °C, 12 h). **c** Shear rheology profiles (G′) at 303 K before (● **A₂B₂**) and after (▲ **ABA**) the topology transformation.

transformation of the $A_2B_2$ block copolymer from a star-shaped (**A₂B₂**) to a linear (**ABA**) topology was carried out via acetylation of the *sec*-ammonium moiety at the center of the polymer chain.

The increase in the hydrodynamic volume and changes in the ¹H NMR spectrum after acetylation supported the star-to-linear topology transformation (Supplementary Figs. 20 and 21). DSC measurements of both copolymers (Supplementary Fig. 22) showed $T_g$ derived from PmVL (–40 °C) and PS blocks, albeit that the $T_g$ of PS block in the linear polymer (50 °C) was lower than that of the star-shaped one (80 °C). The temperature-dependent viscoelastic measurements (Supplementary Fig. 26) indicated that $T_g$ values of PS blocks in **A₂B₂** and **ABA** were ca. 80 and 50 °C, respectively, which were consistent with DSC results. Fig. 6 shows the SAXS profiles before (Fig. 6a) and after (Fig. 6b) the topology transformation. The observed scattering peaks of the **A₂B₂** star polymer indicate that it forms a lamellar structure, while the broad scattering peaks of the **ABA** linear polymer indicate a less-ordered structure (Supplementary Fig. 23). These results are consistent with the TEM images (Supplementary Fig. 24). We assume that the reasons for the less-ordered structure in linear **ABA** are (i) weaker phase segregation and (ii) weaker degree of block junction pinning at the domain interface, as compared to star **A₂B₂**. (i) For the triblock copolymer $A_nB_{2m}A_n$ ($n$, $m$: degree of polymerization), the order–disorder transition temperature is much lower than that of the diblock copolymer $A_{2n}B_{2m}$ (the same molecular weight as $A_nB_{2m}A_n$) and similar to that of $A_nB_m$ (a half molecular weight of $A_nB_{2m}A_n$)[28]. In addition, the phase segregation of star-shaped $(A_n)_2(B_m)_2$ is stronger than that of $A_nB_m$[29]. Therefore, it is deduced from these facts that $A_nB_{2m}A_n$ (**ABA** in this study) is in the weaker segregation state, as compared to star $(A_n)_2(B_m)_2$ (**A₂B₂** in this study). (ii) Block junctions of star **A₂B₂** are expected to be more strongly pinned at the domain interface as compared to linear **AB**, because more severe thermodynamic and topological constraints should act on star **A₂B₂** than linear **AB**[29]. A similar interpretation would be valid for star **A₂B₂** and linear **ABA** (i.e., stronger block junction pinning of star **A₂B₂** than linear **ABA**), because two PS blocks (**A**) in linear **ABA** are apart from each other by the intermediate PmVL block (**B**) and hence should be less correlated with each other. It is deduced from these two reasons that phase segregation becomes weak and domain boundary becomes broad by the topology transformation. Judging from SAXS (Fig. 6a, b) and TEM (Supplementary Fig. 24), linear **ABA** is speculated to form a less-ordered

morphology with broad domain boundary (Supplementary Fig. 23). The decrease in $T_g$ of linear **ABA** compared to star-shaped **A₂B₂** should be attributed to the broadening of domain boundary by the topology transformation (i.e., partial mixing of PmVL and PS blocks in broad domain boundary).

These results indicate that the difference in topology should be responsible for the ordered structure in the multicomponent system, since both copolymers have the same polymer composition. The frequency-dependent viscoelastic measurements (Fig. 6c) revealed the change in mechanical properties, i.e., the storage modulus (G′) value increased after the topology transformation despite the less-ordered structure of **ABA** and the fact that its $T_g$ was lower than that of star-shaped **A₂B₂**. This increase in G′ would be attributed to the cross-linking effect of the microphase-separated structure consisting of hard and soft segments (bridged structure; in Fig. 1b). TPE characteristics were successfully induced by the transformation from the $A_2B_2$ star topology to the ABA linear topology.

## Discussion
Two types of block copolymers whose topology can be transformed via a movable rotaxane linkage at their junction point were synthesized. Such block copolymers represent a system in which the topology transformation of the polymer changes the microphase-separated structure and macroscopic mechanical properties. It is feasible to expect that the concept shown here will be applicable to a wide range of polymers and rotaxane units[30–37]. Therefore, we are convinced that the results of the present study will help to develop the research avenues to the stimuli-responsive systems based on polymer topology transformations. To achieve macroscopic property changes via external stimuli, usually bottom-up approaches are used, which involve accumulation of stimuli-responsive small molecules (e.g., azobenzene-unit-containing systems or rotaxane switching units). While these systems aim to translate nanoscopic molecular phenomena into macroscopic changes[38,39], the macroscopic properties of macromolecules composed of accumulated stimuli-responsive small molecules are generally difficult to predict and are mainly governed by the stimuli-responsive parts. Therefore, the functionality derived from the stimuli-responsive molecules does not necessarily translate into the desired changes in the mechanical properties in the accumulated state in the bulk, which limits the design of materials with specific functions. In the present system, the long-range switching of the rotaxane structure at the junction

points governs the macroscopic properties and functions without any accumulation of functional units. Considering that the variety of potential polymer chains is virtually infinite, the concept of topology transformation via rotaxane linkage could provide access to a new research area in polymer and supramolecular chemistry.

## Data availability

The authors declare that the main data supporting the findings of this study are available within the article and its Supplementary Information files. Extra data are available from the corresponding author upon reasonable request.

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

## Acknowledgements

This work was supported by KAKENHI grants 16H00754 (T.T.) and 18H02018 (T.T.) from the Japan Society for the Promotion of Science (JSPS), and by JST, CREST (JPMJCR1522), Japan. Synchrotron SAXS and WAXD measurements have been performed under the approval of Photon Factory Advisory Committee (Nos. 2016G652 and 2017G084).

## Author contributions

H.S., D.A., S.U., and T.T. conceived and designed the experiments. H.M., S.N., and T.H. contributed to the characterization of polymers by SAXS analysis. X.L. and K.N. contributed to the characterization of polymers by shear rheology analysis. H.S. and D.A. performed the experiments. H.S. analyzed the data. D.A., H.M., and T.T. wrote the manuscript. H.S., D.A., and H.M. contributed equally to this work.

## Competing interests

The authors declare no competing interests.
