## [Peer Review File · Nature Communications]

Topology-transformable block copolymers based on a rotaxane structure: change in bulk properties with same compositionReviewers' comments:

Reviewer #1 (Remarks to the Author):

This paper reports on the topology transformation of copolymers bearing rotaxane junction. A ABCstar to ABClinear transformation is first illustrated, and the impact on the formed solid state morphology is shown. Secondly, a A2B2 star to ABA linear topology is reported together with its influence of the rheology. The paper is interesting but some points are insufficiently discussed and some issues need to be addressed (see below). Moreover, the paper does not contain enough novelty to warrant a publication in Nature Communications. Indeed, the authors already reported several times the concept of topological transformation in polymers using the same chemistry (ref 12,13 14-17 of the present paper). The impact of this change of topology on different properties (change of hydrodynamic radius, impact on viscosity, impact on crystallization, impact on nanoscale morphology of films of block copolymers) was also already illustrated. Even the transformation from a ABC star to a ABC linear copolymer was already reported by the authors (ACS Macro Lett. 2016, 5, 699). Considering this, I cannot recommend this paper for publication in Nature Communications, it is better suited to a more specialised journal.

- The peak integrations, as coming from the used software, should be indicated on all NMR spectra

- The A polymer was purified by preparative GPC. Why was that needed since the polymer is grown from initiator axle 1? Same question for the ABC linear copolymer, why is it needed to purify this compound in such a way after acetylation of the amine group? Especially because it is said in the manuscript that the reaction is quantitative.

- The click reaction to graft the PDMS block seems largely inefficient. Despite quite extreme conditions for this type of reaction (heating, large amount of catalyst used, long reaction time) the yield is very poor (27%). This lack of reactivity should be commented.

- The H NMR peak of the amide group at the extremity of the PVL chain becomes extremely broad (almost not distinguishable from the background) when the ring is around it for the ABC linear copolymer (Figure S6), while it remains sharp for the ABA copolymer (Figure S19). Why such a difference?

- The SAXS data should be discussed in more details:

Why is there a shift to low q when temperature is increasing?

For ABCstar the overall SAXS features are similar with increasing temperature, but for ABClinear there are important changes with temperature, way beyond a simple shift to low q (new peaks are appearing, not all the peaks shift in the same way,...). The origin of this behaviour should be explained.

Is the topology of the copolymer really the same at the different temperature? It could be expected that the mobility of the ring changes with temperature (depending also on the mobility of the polymer chains themselves) as well as its affinity for the amide binding station.

For ABClinear why is there a mixture of morphologies (lamellae and cylinders)? Can it be related to the above comment, i.e. a mixture of copolymer topology?

- Why is there a decrease of PDI after topology transformation for the A2B2 star?

- For the A2B2 star, it is not clear if 17800 g/mol is the Mn of each PS chain or the total Mn for both PS chain.

- The DSC results should be better discussed:

A Tg of 50°C for PS as measured for the ABAlinear is very very low for PS. The authors explain this due to the disordered structure formed by this copolymer, but what is the relation between both points? Styrene is amorphous in both cases and Tg deals with local motions. How can a disordered structure induce a change in Tg (even more, a large 30°C decrease!)? Moreover, why is there no Tg

change for PmVL? It should be affected in a similar way to PS. Is the Tg of PmVL also much lower than expected?

It is also strange that the Tg of PS is barely visible since the fraction of both polymer (PmVL and PS) are very close. This should be explained. DMTA (not shear rheology as the authors have done) should be performed on both copolymer because it is much more sensitive to Tg and other transitions than DSC.

- What is the average spacing deduced from the peak in SAXS for the ABAlinear (Fig 3b)? It indicates indeed an ill-defined structure but a peak is still clearly observed showing some kind of weak ordering or clustering/aggregation with a characteristic spacing. This should be further discussed.

- The data of figure 3c have been obtained by shear rheology, this name should be used instead of DMA.

- The temperature at which the rheology measurements have been done should be indicated, and the G" data should be added in Figure 3c. Why is the G' decreasing with frequency and not keeping constant as one could expect for a crosslinked structure? This indicates some kind of slow and progressive relaxation but where does it come from? Is the difference in rheological property between both topologies the same whatever the temperature?

Reviewer #2 (Remarks to the Author):

The manuscript entitled Topology-transformable block copolymers based on a rotaxane structure by Hiroki Sato and coworkers seems interesting. The experiment is carefully made and the manuscript is well-written. I recommend its publication after the followings being improved.

1. On the removal of the RAFT moieties by AIBN and peroxide, the terminal should be confirmed. Can the RAFT moieties be converted in SH? If this happens, it will exert great influence on the next synthesis of star polymers. The authors should afford suitable characterization to support their declaration.

2. Figure 2. On the topology transformation from ABCstar to ABClinear, DMA of the polymers before and after topology transformation should be made.

3. From the SAXS profiles and bright-field STEM images before and after the topology transformation, morphology difference is observed obviously. However, the deionization, e.g, from n+ to NH, will seriously affect the polymer morphology. The authors should provide information to clarify the exact reason on the morphology difference. This is essential.

Reviewer #3 (Remarks to the Author):

This is an interesting report on topology-transformable block copolymers. The authors synthesized two types of topology-transformable block copolymers using precision synthesis techniques, and they have further demonstrated microphase-separated morphology transitions and mechanical property changes through topology transformation of rotaxane-based block copolymers. Although topology transformation in the solution state has been confirmed so far, the topology transformation in bulk, which is more important for applications to structural or functional materials, has not been explored because of difficulty in synthesis and evaluations. This report is the first report on the effect of topology transformation of rotaxane-based block copolymer in bulk, and the study has clearly a high level of novelty, also exhibiting scientific significance. Therefore, I recommend publication of the manuscript with minor revisions noted below.

Minor comments:

1. In the Introduction section, polyurethane was shown as an example of how the microphase-separated structure affects macroscopic mechanical properties. But since the subject of this study is a

block copolymer rather than polyurethane, the authors should mention the mechanical property differences between ABA triblock copolymers and AB diblock copolymers.

2. Figure 1a shows a schematic illustration of cylindrical structure for ABCstar. On the basis of experimental results, the authors concluded that it was a spherical structure. It is better to show a schematic illustration of a spherical structure in Figure 1a.

3. One of the important points of this study is the precision synthesis of the rotaxane-based block copolymers. GPC chromatograms guarantee the precision synthesis and are also important data to back up the topology transformation. Can the authors incorporate Figures S5 and S20 into Figures 2 and 3, respectively?

4. Are the SAXS profiles in Figure 2 measured at 70 ° C? Microphase-separated structures close to equilibrium are formed when samples are annealed. Why didn't the authors show profiles of 120 ° C which is higher than the polystyrene Tg? The SAXS profiles at 120 °C in Figures S14 and S16 seem to be a better agreement with the conclusions of authors' structural analysis.

5. The fact that the Tg of polystyrene in ABA is 50 °C is attributed to the disordered state, whereas the reason for the disordered state observed by STEM and SAXS is not explained at all. It might be useful for the authors to know the Matsen's report on molecular architecture affecting the phase behavior (Macromolecules 2012, 45, 2161-2165.).

6. If A2B2 does not have physical cross-links while ABA has physical cross-links, it is better to compare A2B2 and ABA in terms of strain robustness rather than the absolute value of elastic modulus. If the polymer network is formed by physical cross-links, linear viscoelasticity can be observed even when a larger strain is applied; conversely, if there is no physical cross-link, non-linear viscoelasticity is easily observed by applying a smaller strain. Strain amplitude sweeps will probably reveal better strain robustness of ABA triblock than A2B2 tetrablock.

7. Is "Alas" on the second line in Abstract a mistake?

Response to the comments of reviewer 1:

The paper is interesting but some points are insufficiently discussed and some issues need to be addressed (see below). Moreover, the paper does not contain enough novelty to warrant a publication in Nature Communications. Indeed, the authors already reported several times the concept of topological transformation in polymers using the same chemistry (ref 12,13 14-17 of the present paper). The impact of this change of topology on different properties (change of hydrodynamic radius, impact on viscosity, impact on crystallization, impact on nanoscale morphology of films of block copolymers) was also already illustrated. Even the transformation from a ABC star to a ABC linear copolymer was already reported by the authors (ACS Macro Lett. 2016, 5, 699). Considering this, I cannot recommend this paper for publication in Nature Communications, it is better suited to a more specialised journal.

We wish to express our appreciation to the reviewer for his or her insightful comments, which have helped us significantly improve the paper.

First of all, the novelty of this research is change in bulk properties of polymer, 1) microphase-separated structures and 2) macroscopic mechanical properties, via topology transformation with rotaxane structure. As the reviewer suggested, we have already reported topology transformation of polymers and synthetic method of ABC polymer. However, those reports only show change in “solution state”, not in bulk state. The polymers are usually used in bulk state as the materials and change in microphase-separated structures and macroscopic mechanical properties would show great potential for not only polymer chemistry but also supramolecular chemistry. This is the first report showing change in 1) microphase-separated structures and 2) macroscopic mechanical properties, via topology transformation of block copolymers with rotaxane structure. To emphasize our originality of this research, the title was revised from “*Topology-transformable block copolymers based on a rotaxane structure*” to “*Topology-transformable block copolymers based on a rotaxane structure: change in bulk properties with same polymer composition*”

- *The peak integrations, as coming from the used software, should be indicated on all NMR spectra*

We have added NMR spectra with peak integrations in SI, as the reviewer suggested.

- *The A polymer was purified by preparative GPC. Why was that needed since the polymer is grown from initiator axle 1? Same question for the ABC linear copolymer, why is it needed to purify this compound in such a way after acetylation of the amine group? Especially because it is said in the manuscript that the reaction is quantitative.*

In the synthesis of A polymer, preparative GPC was used to remove excess wheel components which work as the initiators in the synthesis of AB block copolymer. In other steps, just in case, i.e., we have used preparative GPC to completely remove small compounds like triethylamine, acetic anhydride, and solvent which affect

NMR spectrum, and to get block copolymers without contaminations.

- The click reaction to graft the PDMS block seems largely inefficient. Despite quite extreme conditions for this type of reaction (heating, large amount of catalyst used, long reaction time) the yield is very poor (27%). This lack of reactivity should be commented.

As the reviewer suggested, the isolated yield of copper-catalyzed alkyne – azide cycloaddition was relatively low. Steric hindrance around alkyne group on the axle polymer might result in less reactivity. The purification process using preparative GPC to obtain perfectly pure ABC terpolymer also affect the decrease in yield. We have added comment about low yield in manuscript.

- The ¹H NMR peak of the amide group at the extremity of the PVL chain becomes extremely broad (almost not distinguishable from the background) when the ring is around it for the ABC linear copolymer (Figure S6), while it remains sharp for the ABA copolymer (Figure S19). Why such a difference?

We assume that a wheel component localized around NH group but still moving, i.e., not fixed perfectly like covalent bonds. The difference in polymer structure of PVL and PmVL might affect localized state of wheel component. We assumed that the bulkiness of PmVL with methyl group on axle chain works as small stopper (ref. Angew. Chem. Int. Ed. 2019, 58, 2765-2768) and promote the localization on NH group, resulting in the difference of NMR peak of the amide group.

- The SAXS data should be discussed in more details:

Why is there a shift to low q when temperature is increasing?

For ABCstar the overall SAXS features are similar with increasing temperature, but for ABClinear there are important changes with temperature, way beyond a simple shift to low q (new peaks are appearing, not all the peaks shift in the same way, ...). The origin of this behaviour should be explained.

Is the topology of the copolymer really the same at the different temperature? It could be expected that the mobility of the ring changes with temperature (depending also on the mobility of the polymer chains themselves) as well as its affinity for the amide binding station.

For ABClinear why is there a mixture of morphologies (lamellae and cylinders)? Can it be related to the above comment, i.e. a mixture of copolymer topology?

Thank you for essential comments. As the reviewer suggested, the peak shift to lower q was observed at 120 °C in both ABC polymers. At 70 °C, the PS domain is in the glassy state, whereas PDMS and PVL domains are in the molten state. Therefore, the peak shift to lower q should be attributed to the glass transition of PS [70 °C < T_g (85–95 °C) < 120 °C]. The significantly increased mobility of PS domains in both ABC polymers should lead to a more ordered morphology and hence more clear scattering peaks at 120 °C (especially in ABC_{linear}). At the same time, thermal expansion of rubbery PS domains should cause an increase in domain spacings (i.e., peak shift to lower q).

Because the interaction between urethane linkage/crown ether should be weakened and the mobility of wheel

component would be increased, topology might be affected by temperature, especially at 120 °C. We used SAXS profiles at 70 ° C in the manuscript, because the peak intensities increased (especially in **ABC_{linear}**) while the primary peak positions were almost the same with increasing temperature to 70 ° C (Figures S11-S12), which makes the discussion simple and is comparable to the STEM image.

We have added the sentence about the shift to lower q and the comment on that in the manuscript as follows; “The peak shift to lower q was observed at 120 °C in both **ABC** polymers. At 70 ° C, the PS domain is in the glassy state, whereas PDMS and PVL domains are in the molten state. Therefore, the peak shift to lower q should be attributed to the glass transition of PS [$70\text{ °C} < T_g (85\text{--}95\text{ °C}) < 120\text{ °C}$]. The significantly increased mobility of PS domains in both **ABC** polymers should lead to a more ordered morphology, and hence more clear scattering peaks at 120 °C (especially in **ABC_{linear}**). At the same time, thermal expansion of rubbery PS domains should cause an increase in domain spacings (i.e., peak shift to lower q). Because the interaction between urethane linkage/crown ether should be weakened and the mobility of wheel component would be increased, topology might be affected by temperature, especially at 120 °C, which makes an effect on morphology of **ABC_{linear}** but the discussion complicated when comparing SAXS profiles and STEM images. Therefore, the SAXS profiles at 70 ° C are shown as representative, because the peak intensities increased (especially in **ABC_{linear}**) while the primary peak positions were almost the same with increasing temperature to 70 ° C (Figures S11-S12). “

As for the mixture of morphologies (lamellae and cylinders), we have not concluded at this point. However, we do insist that topology transformation in this report clearly changed morphologies, which is what we would like to insist the most.

- *Why is there a decrease of PDI after topology transformation for the A2B2 star?*

There is small difference in PDI before and after topology transformation (1.2 vs 1.1). Two sec-ammonium moieties might affect broad GPC but 1.2 is still narrow.

- *For the A2B2 star, it is not clear if 17800 g/mol is the Mn of each PS chain or the total Mn for both PS chain. 17800 g/mol is total Mn. We have added comment in manuscript “(PS; total Mn for both PS chains)”.*

- *The DSC results should be better discussed:*

A T_g of 50° C for PS as measured for the ABAlinear is very very low for PS. The authors explain this due to the disordered structure formed by this copolymer, but what is the relation between both points? Styrene is amorphous in both cases and T_g deals with local motions. How can a disordered structure induce a change in T_g (even more, a large 30° C decrease!)? Moreover, why is there no T_g change for PmVL? It should be affected in a similar way to PS. Is the T_g of PmVL also much lower than expected?

It is also strange that the T_g of PS is barely visible since the fraction of both polymer (PmVL and PS) are very

close. This should be explained. DMTA (not shear rheology as the authors have done) should be performed on both copolymer because it is much more sensitive to T_g and other transitions than DSC.

As the reviewer mentioned we have explained this due to the disordered structure formed by this copolymer in manuscript. As change in specific heat capacity of PMeVL is much higher than that of PS, change in T_g of PMeVL was not observed remarkably, while T_g of PS was observed remarkably. We also think the small molecular weight of PS (8900 g/mol each PS chain), which is less than 8900 g/mol, also affect the result. In this system, we used PMeVL as the soft segment and the molecular weight of PMeVL is 18600. Hopefully, the small volume of soft segment (PMeVL) to hard segments (PS) would be ideal as the thermo plastic elastomer. Therefore, molecular weight of PS in this study is relatively small, or mechanical properties govern by PS, i.e., change in property before and after topology transformation would be small. In this research, we would like to show novel concept that macroscopic mechanical properties can be changed via topology transformation of block copolymers with rotaxane structure. We have tried to use DMTA but the sample is not suit for DMTA, that why we used shear rheology in this research.

- *What is the average spacing deduced from the peak in SAXS for the ABA linear (Fig 3b)? It indicates indeed an ill-defined structure but a peak is still clearly observed showing some kind of weak ordering or clustering/aggregation with a characteristic spacing. This should be further discussed.*

Thank you for valuable comments. The average spacing deduced from the SAXS peak of linear ABA is 16.5 nm ($q = 0.38 \text{ nm}^{-1}$). We assumed that the reasons for the less-ordered structure in linear ABA are (i) *weaker phase segregation* and (ii) *weaker degree of block junction pinning at domain interface*, as compared to star A_2B_2 . (i) For triblock copolymer $A_nB_{2m}A_n$ (n, m : degree of polymerization), the order–disorder transition temperature is much lower than that of diblock copolymer $A_{2n}B_{2m}$ (comparable molecular weight to $A_nB_{2m}A_n$) and similar to that of A_nB_m (a half molecular weight of $A_nB_{2m}A_n$) (*Macromolecules* **2000**, *33*(14), 5124–5130). In addition, phase segregation of star-shaped $(A_n)_2(B_m)_2$ is stronger than that of A_nB_m (*Macromolecules* **2000**, *33*(22), 8399–8414.). Therefore, it is deduced from these facts that $A_nB_{2m}A_n$ (ABA in this study) is in the weaker segregation state, as compared to star $(A_n)_2(B_m)_2$ (A_2B_2 in this study). (ii) Block junctions of star A_2B_2 are expected to be more strongly pinned at domain interface as compared to linear AB, because more severe thermodynamic and topological constraints should act on star A_2B_2 than linear AB (*Macromolecules* **2000**, *33*, 22, 8399–8414). A similar interpretation would be valid for star A_2B_2 and linear ABA (i.e., stronger block junction pinning of star A_2B_2 than linear ABA), because two PS blocks (A) in linear ABA are apart from each other by the intermediate PmVL block (B) and hence should be less correlated with each other. It is deduced from these two reasons that phase segregation becomes weak and domain boundary becomes broad by topology transformation. Judging from SAXS (Figure 3) and TEM (Figure S23), linear ABA is speculated to form a less-ordered morphology with broad domain boundary. The decrease in T_g of linear ABA compared to star-shaped A_2B_2 should be attributed to broadening of domain boundary by topology transformation.

We have added discussion on the morphology change by topology transformation from A_2B_2 to ABA in the

manuscript as follows;

We assumed that the reasons for the less-ordered structure in linear **ABA** are (i) *weaker phase segregation* and (ii) *weaker degree of block junction pinning at domain interface*, as compared to star **A₂B₂**. (i) For triblock copolymer **A_nB_{2m}A_n** (*n, m*: degree of polymerization), the order–disorder transition temperature is much lower than that of diblock copolymer **A_{2n}B_{2m}** (comparable molecular weight to **A_nB_{2m}A_n**) and similar to that of **A_nB_m** (a half molecular weight of **A_nB_{2m}A_n**) (*Macromolecules* **2000**, *33* (14), 5124–5130). In addition, phase segregation of star-shaped (**A_n)₂(B_m)₂** is stronger than that of **A_nB_m** (*Macromolecules* **2000**, *33* (22), 8399–8414.). Therefore, it is deduced from these facts that **A_nB_{2m}A_n** (**ABA** in this study) is in the weaker segregation state, as compared to star (**A_n)₂(B_m)₂** (**A₂B₂** in this study). (ii) Block junctions of star **A₂B₂** are expected to be more strongly pinned at domain interface as compared to linear **AB**, because more severe thermodynamic and topological constraints should act on star **A₂B₂** than linear **AB** (*Macromolecules* **2000**, *33*, 22, 8399–8414). A similar interpretation would be valid for star **A₂B₂** and linear **ABA** (i.e., stronger block junction pinning of star **A₂B₂** than linear **ABA**), because two PS blocks (**A**) in linear **ABA** are apart from each other by the intermediate PmVL block (**B**) and hence should be less correlated with each other. It is deduced from these two reasons that phase segregation becomes weak and domain boundary becomes broad by topology transformation. Judging from SAXS (Figure 3) and TEM (Figure S23, linear **ABA** is speculated to form a less-ordered morphology with broad domain boundary. The decrease in *T_g* of linear **ABA** compared to star-shaped **A₂B₂** should be attributed to broadening of domain boundary by topology transformation.

- *The data of figure 3c have been obtained by shear rheology, this name should be used instead of DMA.*

We have revised DMA to shear rheology as the reviewer suggested.

- *The temperature at which the rheology measurements have been done should be indicated, and the G'' data should be added in Figure 3c. Why is the G' decreasing with frequency and not keeping constant as one could expect for a crosslinked structure? This indicates some kind of slow and progressive relaxation but where does it come from? Is the difference in rheological property between both topologies the same whatever the temperature?*

We have added G'' data in supporting information and commented on manuscript. The reason of decrease in G' is that this is not fully cross-linked polymer like chemical cross-linked polymer, but thermoplastic elastomer (ref. *Polymer* **2018**, *10*, 400). Rheology measurements were done at same temperature and same day.

Response to the comments of reviewer 2:

The manuscript entitled Topology-transformable block copolymers based on a rotaxane structure by Hiroki Sato and coworkers seems interesting. The experiment is carefully made and the manuscript is well-written. I recommend its publication after the followings being improved.

We wish to express our appreciation to the reviewer for his or her insightful comments, which have helped us

significantly improve the paper.

1. On the removal of the RAFT moieties by AIBN and peroxide, the terminal should be confirmed. Can the RAFT moieties be converted in SH? If this happens, it will exert great influence on the next synthesis of star polymers. The authors should afford suitable characterization to support their declaration.

As the author suggested, RAFT moiety might affect the conversion of click reaction. That is why we have applied removal of the RAFT moieties by AIBN and peroxide. Although it was quite difficult to confirm the removal of RAFT moieties by NMR, we have confirmed it using model polymer (ref. *J. Polym. Chem.* 2014, 5, 2461-2471).

2. Figure 2. On the topology transformation from ABCstar to ABClinear, DMA of the polymers before and after topology transformation should be made.

ABC polymer is designed to confirm topology change visually, i.e., combination of TEM and SAXS. Unfortunately, the properties of them including crystalline polymer (PVL), are not ideal for DMA measurements. That is why we designed A2B2 polymers as an alternative, i.e., A and B components are both amorphous and viscous character which are suitable for DMA (Shear rheology) measurements.

3. From the SAXS profiles and bright-field STEM images before and after the topology transformation, morphology difference is observed obviously. However, the deionization, e.g, from n+ to NH, will seriously affect the polymer morphology. The authors should provide information to clarify the exact reason on the morphology difference. This is essential.

Majority of components is ABC polymers and ratio of sec-ammonium moiety (N-acetyl group) is quite small, which would not affect polymer morphology. Furthermore, sec-ammonium moiety is surrounded by the crown ether, which acts as naturalized molecules. Topology transformation, which is supported by change in GPC profiles, is driving force of change in morphology as we designed.

Response to the comments of reviewer 3:

Although topology transformation in the solution state has been confirmed so far, the topology transformation in bulk, which is more important for applications to structural or functional materials, has not been explored because of difficulty in synthesis and evaluations. This report is the first report on the effect of topology transformation of rotaxane-based block copolymer in bulk, and the study has clearly a high level of novelty, also exhibiting scientific significance. Therefore, I recommend publication of the manuscript with minor revisions noted below.

We wish to express our appreciation to the reviewer for his or her insightful comments, which have helped us significantly improve the paper.

1. In the Introduction section, polyurethane was shown as an example of how the microphase-separated

structure affects macroscopic mechanical properties. But since the subject of this study is a block copolymer rather than polyurethane, the authors should mention the mechanical property differences between ABA triblock copolymers and AB diblock copolymers.

As the reviewer suggested, we revised the introduction part, i.e., describing mechanical property differences between ABA triblock copolymers and AB diblock copolymers.

2. Figure 1a shows a schematic illustration of cylindrical structure for ABC_{star}. On the basis of experimental results, the authors concluded that it was a spherical structure. It is better to show a schematic illustration of a spherical structure in Figure 1a.

Thank you for your great advice, but we have not revealed the detail of spherical structure yet, i.e., PDMS shows high resolution but the difference in PS and PVL is not clear). Figure 1 is described as 2D and it can be regarded as spherical structure if the two colors resolution is low (regarded as same color). If Figure 1a is not appropriate, we would like to remove the cartoon.

3. One of the important points of this study is the precision synthesis of the rotaxane-based block copolymers. GPC chromatograms guarantee the precision synthesis and are also important data to back up the topology transformation. Can the authors incorporate Figures S5 and S20 into Figures 2 and 3, respectively?

We have already reported the precision synthesis of ABC block polymer and change in hydrodynamic volume (ACS Macro Lett. 5, 699-703). The novelty of this research is change in bulk properties of polymer, 1) microphase-separated structures and 2) macroscopic mechanical properties, via topology transformation with rotaxane structure. Although the change in hydrodynamic volume demonstrate the precision synthesis and topology change, minimum required data are described in this communication. Thank you for your suggestion.

4. Are the SAXS profiles in Figure 2 measured at 70 ° C? Microphase-separated structures close to equilibrium are formed when samples are annealed. Why didn't the authors show profiles of 120 ° C which is higher than the polystyrene T_g? The SAXS profiles at 120 ° C in Figures S14 and S16 seem to be a better agreement with the conclusions of authors' structural analysis.

As the reviewer suggested, we have confirmed clear peaks at 120 °C in SR-SAXS profiles of **ABC_{linear}**. We used profiles at 70 ° C in manuscript, because the peak intensities increased (especially in **ABC_{linear}**) while the primary peak positions were almost the same with increasing temperature to 70 ° C (Figures S11-S12), which make the discussion simple and comparable to the STEM image. Actually, the spacings calculated from the primary peak positions (26 nm for **ABC_{star}**; 37 nm for lamella structure in **ABC_{linear}**) are in good agreement with those obtained from STEM images (24 nm for **ABC_{star}**; 37 nm for lamellar structure in **ABC_{linear}**).

5. The fact that the T_g of polystyrene in ABA is 50 ° C is attributed to the disordered state, whereas the reason for the disordered state observed by STEM and SAXS is not explained at all. It might be useful for the authors to know the Matsen's report on molecular architecture affecting the phase behavior (Macromolecules

Thank you for your great suggestion. Similar discussion was made on the comment to reviewer 1 as decried in following part.

We assumed that the reasons for the less-ordered structure in linear **ABA** are (i) *weaker phase segregation* and (ii) *weaker degree of block junction pinning at domain interface*, as compared to star **A₂B₂**. (i) For triblock copolymer **A_nB_{2m}A_n** (n, m : degree of polymerization), the order–disorder transition temperature is much lower than that of diblock copolymer **A_{2n}B_{2m}** (comparable molecular weight to **A_nB_{2m}A_n**) and similar to that of **A_nB_m** (a half molecular weight of **A_nB_{2m}A_n**) (*Macromolecules* **2000**, *33* (14), 5124–5130). In addition, phase segregation of star-shaped **(A_n)₂(B_m)₂** is stronger than that of **A_nB_m** (*Macromolecules* **2000**, *33* (22), 8399–8414.). Therefore, it is deduced from these facts that **A_nB_{2m}A_n** (**ABA** in this study) is in the weaker segregation state, as compared to star **(A_n)₂(B_m)₂** (**A₂B₂** in this study). (ii) Block junctions of star **A₂B₂** are expected to be more strongly pinned at domain interface as compared to linear **AB**, because more severe thermodynamic and topological constraints should act on star **A₂B₂** than linear **AB** (*Macromolecules* **2000**, *33*, 22, 8399–8414). A similar interpretation would be valid for star **A₂B₂** and linear **ABA** (i.e., stronger block junction pinning of star **A₂B₂** than linear **ABA**), because two PS blocks (**A**) in linear **ABA** are apart from each other by the intermediate PmVL block (**B**) and hence should be less correlated with each other. It is deduced from these two reasons that phase segregation becomes weak and domain boundary becomes broad by topology transformation. Judging from SAXS (Figure 3) and TEM (Figure S23), linear **ABA** is speculated to form a less-ordered morphology with broad domain boundary. The decrease in T_g of linear **ABA** compared to star-shaped **A₂B₂** should be attributed to broadening of domain boundary by topology transformation.

We have added discussion on the morphology change by topology transformation from **A₂B₂** to **ABA** in the manuscript as follows;

We assumed that the reasons for the less-ordered structure in linear **ABA** are (i) *weaker phase segregation* and (ii) *weaker degree of block junction pinning at domain interface*, as compared to star **A₂B₂**. (i) For triblock copolymer **A_nB_{2m}A_n** (n, m : degree of polymerization), the order–disorder transition temperature is much lower than that of diblock copolymer **A_{2n}B_{2m}** (comparable molecular weight to **A_nB_{2m}A_n**) and similar to that of **A_nB_m** (a half molecular weight of **A_nB_{2m}A_n**) (*Macromolecules* **2000**, *33* (14), 5124–5130). In addition, phase segregation of star-shaped **(A_n)₂(B_m)₂** is stronger than that of **A_nB_m** (*Macromolecules* **2000**, *33* (22), 8399–8414.). Therefore, it is deduced from these facts that **A_nB_{2m}A_n** (**ABA** in this study) is in the weaker segregation state, as compared to star **(A_n)₂(B_m)₂** (**A₂B₂** in this study). (ii) Block junctions of star **A₂B₂** are expected to be more strongly pinned at domain interface as compared to linear **AB**, because more severe thermodynamic and topological constraints should act on star **A₂B₂** than linear **AB** (*Macromolecules* **2000**, *33*, 22, 8399–8414). A similar interpretation would be valid for star **A₂B₂** and linear **ABA** (i.e., stronger block junction pinning of star **A₂B₂** than linear **ABA**), because two PS blocks (**A**) in linear **ABA** are apart from each other by the intermediate PmVL block (**B**) and hence should be less correlated with each other. It is deduced from these two reasons

that phase segregation becomes weak and domain boundary becomes broad by topology transformation. Judging from SAXS (Figure 3) and TEM (Figure S23, linear **ABA** is speculated to form a less-ordered morphology with broad domain boundary. The decrease in T_g of linear **ABA** compared to star-shaped **A₂B₂** should be attributed to broadening of domain boundary by topology transformation.

6. If A2B2 does not have physical cross-links while ABA has physical cross-links, it is better to compare A2B2 and ABA in terms of strain robustness rather than the absolute value of elastic modulus. If the polymer network is formed by physical cross-links, linear viscoelasticity can be observed even when a larger strain is applied; conversely, if there is no physical cross-link, non-linear viscoelasticity is easily observed by applying a smaller strain. Strain amplitude sweeps will probably reveal better strain robustness of ABA triblock than A2B2 tetrablock.

Thank you for your great advice. Unfortunately, we don't have enough sample to check strain amplitude sweeps. We are going to check the strain amplitude sweeps with different polymers and summarized as full paper in near future.

7. Is "Alas" on the second line in Abstract a mistake?

We have used "However" instead of "Alas".

Reviewers' comments:

Reviewer #1 (Remarks to the Author):

The authors have answered to some of my previous comments but there are still some issues that need to be addressed. Moreover, they say several times in their reply that their aim is to report the effect of the topology transformation on the bulk properties. I understand that, but it is still needed to understand the different results to back up the main message and to be sure of the origin of the observed behaviors. Focusing only on results that fit the main message and discarding the others as being not important or minor points, is not the way to proceed to my opinion.

- The integrations on the NMR spectrum of Fig S1-2 (added in the revised version) show that the A polymer is not pure and contains an excess (almost 2-fold) of the PVL axle. Was this excess removed at one point? If not, the final ABC copolymer is contaminated with AC diblock, which should have an influence on the phase separation.

- In the caption of the new Fig S2-2, it is written "The polymer composition (ratio of A and B) is not same as ABC star polymers, i.e., 12600(PVL):23100(PS):10000(PDMS), because synthesis of ABC star polymers went through purification process which changes polymer composition." I don't understand that. The A/B ratio should be the same in the AB diblock and in the ABC triblock since the C block is grafted by CuAAC on the AB diblock. This should not affect the A/B ratio.

- My previous comment " For ABCstar the overall SAXS features are similar with increasing temperature, but for ABClinear there are important changes with temperature, way beyond a simple shift to low q (new peaks are appearing, not all the peaks shift in the same way,...). The origin of this behaviour should be explained." has not been properly addressed. If the authors want to report on the effect of a topology change on the solid-state morphology, it is important to have a general understanding of the solid state behavior of the two involved copolymers (ABC linear and star).

- Same remark for my comment "For ABClinear why is there a mixture of morphologies (lamellae and cylinders)? Can it be related to the above comment, i.e. a mixture of copolymer topology?". The authors say that they don't conclude on this point, but to me it is important for the global understanding of the underlying mechanisms/phenomena.

- The author's reply about the very low T_g (50°C) measured for PS is not satisfactory. The rather low molar mass of PS (8900 g/mol) is not sufficient to explain this value (see Flory-Fox equation for PS). In fact, by looking at Figure S21, the T_g of the PS block cannot be seen on the DSC graph. The data are noisy and there is no clear evidence of the Cp jump characteristic of the T_g for PS. Hence the value cannot be determined. If the authors want to discuss T_g of this sample, another method to measure it should be used. The authors say that DMTA is not suitable for their sample, but I don't see why. Supported DMTA should work in any case. At the very least, shear rheology at different temperatures below and above the T_g should be provided.

Reviewer #2 (Remarks to the Author):

The revised manuscript is improved. I support its publication in the present form.

Reviewer #3 (Remarks to the Author):

The authors have made a thorough revision to the original manuscript according to all the reviewers' comments; therefore, I recommend publication of the manuscript in Nature Communications as it is.

REPLY TO REVIEWER COMMENTS

Reviewer #1 (Remarks to the Author):

The authors have answered to some of my previous comments but there are still some issues that need to be addressed. Moreover, they say several times in their reply that their aim is to report the effect of the topology transformation on the bulk properties. I understand that, but it is still needed to understand the different results to back up the main message and to be sure of the origin of the observed behaviors. Focusing only on results that fit the main message and discarding the others as being not important or minor points, is not the way to proceed to my opinion.

- The integrations on the NMR spectrum of Fig S1-2 (added in the revised version) show that the A polymer is not pure and contains an excess (almost 2-fold) of the PVL axle. Was this excess removed at one point? If not, the final ABC copolymer is contaminated with AC diblock, which should have an influence on the phase separation.

Thank you for detailed reviewing of our manuscript, but we would like to show a rebuttal for the reviewer's comment. Although some integral ratios are not perfectly consistent with theoretical values because of high molecular mass, the present method actually gives the pure macromolecular [2]rotaxane without any PVL polymer having no wheel component, which was demonstrated by MALDI-TOF-MS in the previous reports (*ACS Macro Lett.*, **2016**, 5, 699-703, *ACS Macro Lett.* 2014, 3, 324-328). There is a possibility of containing original wheel component but it was removed by the preparative GPC, resulting in pure macromolecular [2]rotaxane, A polymer. The usefulness of the present method to synthesize pure macromolecular [2]rotaxane was demonstrated by the previous reports from our group (*Chem. Asian J.* **2018**, 13, 785-789, *ACS Macro Lett.*, **2016**, 5, 699-703, *Chem. Eur. J.*, **2016**, 22, 8759-8762, *ACS Macro Lett.*, **2015**, 4, 598-601, *ACS Macro Lett.* 2014, 3, 324-328, *ACS Macro Lett.*, **2013**, 2, 461-465). Therefore, contamination of AC diblock copolymer hardly occurs in the present system.

- In the caption of the new Fig S2-2, it is written "The polymer composition (ratio of A and B) is not same as ABC star polymers, i.e., 12600(PVL):23100(PS):10000(PDMS), because synthesis of ABC star polymers went through purification process which changes polymer composition." I don't understand that. The A/B ratio should be the same in the AB diblock and in the ABC triblock since the C block is grafted by CuAAC on the AB diblock. This should not affect the A/B ratio.

Thank you for pointing out. To remove an excess amount of PDMS (C) after synthesis of

ABC polymer, preparative GPC was applied several times. This purification process also removed the ABC polymers with relatively low molecular weight, especially whose PS (B polymer) length is short, resulting in the increase of PS fraction in ABC polymers. As the reviewer suggested, we have added the comment as mentioned here in SI.

- My previous comment " For ABCstar the overall SAXS features are similar with increasing temperature, but for ABClinear there are important changes with temperature, way beyond a simple shift to low q (new peaks are appearing, not all the peaks shift in the same way,...). The origin of this behaviour should be explained." has not been properly addressed. If the authors want to report on the effect of a topology change on the solid-state morphology, it is important to have a general understanding of the solid state behavior of the two involved copolymers (ABC linear and star).

As the reviewer suggested, we deeply thought about the solid-state behavior of the present system and then noticed the clear difference of crystallinity originated from PVL block ($\chi_{c,PVL}$) between star and linear topologies, which is supported by DSC (Figure S7) and WAXD (Figure S17, newly added in SI). At room temperature, $\chi_{c,PVL}$ of **ABC_{linear}** (43%) is about 3 times higher than that of **ABC_{star}** (14%). Higher crystallinity should have a larger effect on the solid-state morphology (i.e., competition of crystallization and microphase separation). PS block is hard but PDMS block is soft, so that the crystallization of PVL block can deform the microdomain structure to the extent depending on $\chi_{c,PVL}$. In the case of **ABC_{star}**, SAXS profiles show only slight change with increasing temperature from RT to 70 °C. In contrast, SAXS profiles of **ABC_{linear}** show a drastic change with increasing temperature to 70 °C. Such clear differences in SAXS temperature dependence between star and linear would be attributed to a large difference in their $\chi_{c,PVL}$ values. In the case of **ABC_{star}**, only a small amount of PVL block can crystallize ($\chi_{c,PVL} = 14\%$), so that the microdomain structure should be less affected by the crystallization of PVL block. On the contrary, highly crystalline PVL block in **ABC_{linear}** ($\chi_{c,PVL} = 43\%$) should deform the microdomain structure (mainly soft PDMS domain). The deformed microdomain structure should become ordered at 70 °C, which was detected as a drastic increase in SAXS peak intensity, by melting of PVL crystals ($T_m \sim 50$ °C). Note that PS domain is still in the glassy state at 70 °C ($< T_g$ of PS block), so that structural defects cannot be sufficiently removed. With further increasing temperature to 120 °C ($> T_g$ of PS block), the system can become closer to the equilibrium ordered morphology, which is supported by sharpened and newly-appeared higher-order SAXS peaks.

In addition, the above-mentioned large difference in $\chi_{c,PVL}$ clearly demonstrates that the topology transformation from star to linear much affects the crystallization of constituent block and bulk properties depending on crystallinity (e.g., thermal and mechanical properties).

We have revised the manuscript as mention here and added WAXD data in SI (Figure S17);

“Here, the temperature-dependent changes in ordered morphologies of two ABC terpolymers are discussed based on simultaneous SAXS and wide-angle X-ray diffraction (WAXD, Figure S17) data. It should be noted that the SAXS profile of **ABC_{star}** is clearly different from that of **ABC_{linear}** irrespective of temperature (Figures S11-S16). In the case of **ABC_{star}**, SAXS profiles show only slight change with increasing temperature from room temperature to 70 °C (Figure S11). In contrast, SAXS profiles of **ABC_{linear}** show a drastic change with increasing temperature to 70 °C (Figure S12). This clear difference between **ABC_{star}** and **ABC_{linear}** would be attributed to a large difference in the crystallinity of PVL block ($\chi_{c,PVL}$), which was evaluated from WAXD profiles. In the case of **ABC_{star}**, only a small amount of PVL block can crystallize ($\chi_{c,PVL} = 14\%$), so that the microdomain structure should be less affected by the crystallization of PVL block. On the contrary, highly crystalline PVL blocks in **ABC_{linear}** ($\chi_{c,PVL} = 43\%$) should deform the microdomain structure (mainly soft PDMS domain).²² The drastic increase in SAXS peak intensity of **ABC_{linear}** indicates the ordering of distorted microdomain structure by melting of PVL crystals. Because the PS domain is still in the glassy state at 70 °C [$< T_g$ of PS block (95 °C)], structural defects cannot be sufficiently removed. With further increasing temperature to 120 °C ($> T_g$ of PS block), the system of **ABC_{linear}** would become closer to the equilibrium ordered morphology, which is supported by sharpened and newly-appeared higher-order SAXS peaks (Figures S12, S16). In addition, the peak shift to lower q (increase in domain spacings) was observed at 120 °C in both **ABC** terpolymers, which can be explained by the ordering of microphase-separated structure (i.e., block chains are more stretched) and thermal expansion of rubbery PS domains.”

- Same remark for my comment "For ABC_{linear} why is there a mixture of morphologies (lamellae and cylinders)? Can it be related to the above comment, i.e. a mixture of copolymer topology?". The authors say that they don't conclude on this point, but to me it is important for the global understanding of the underlying mechanisms/phenomena.

As mentioned in the previous part, we repeatedly purified polymers at each process, so that a mixture of copolymers (i.e., star and linear) is improbable. However, a mixture of copolymers with different topologies could be formed in **ABC_{linear}**. Because the interaction between urethane linkage/crown ether in **ABC_{linear}** is not so strong as *sec*-ammonium/crown ether interaction in **ABC_{star}**. Therefore, wheel component could move as temperature increases, resulting in the change in topology depending on the position of wheel component. In most cases, wheel component could be loosely fixed at the end of PVL chain, but probably some amount of wheel is not located at the end of PVL chain, which seems like a mixture of

copolymer topologies. The polymers with similar but different topologies separately could form different morphologies, leading to a mixture of morphologies. Another hypothesis is that this microdomain structure is not a mixture of lamellar and cylinder-like structures but a single morphology, hexagonally perforated layered morphology (HPL). In HPL, lamellar and cylinder structures can be observed depending on viewing directions. Therefore, if some grains with different orientations are contained in the field of view of STEM, it is possible to observe both lamellar and cylinder structures at the same time. This situation might correspond to **ABC_{linear}** (Fig. S9). However, our STEM images are not clear enough to identify the morphology as HPL, probably because the microdomain structure was distorted by the crystallization of PVL block, as suggested by weak SAXS peaks.

Although the detailed structural model of **ABC_{linear}** in the melt remains an open question, we emphasize again that the star-to-linear topology transformation drastically changes the microphase-separated structure.

We have revised the manuscript as follows. “The microphase-separated structure of **ABC_{linear}** can be interpreted by two models; (i) Because the interaction between urethane linkage/crown ether in **ABC_{linear}** is not so strong as *sec*-ammonium/crown ether interaction in **ABC_{star}**. Therefore, the wheel component might move as temperature increases, resulting in the change in topology depending on the position of wheel component. In most cases, the wheel component should be loosely fixed at the end of PVL chain, but probably some amount of wheel is not located at the end of PVL chain, which seems like a mixture of copolymer topologies. The polymers with similar but different topologies might separately form different morphologies, leading to a mixture of morphologies. (ii) Another structural model is a single ordered morphology, a hexagonally perforated layer (HPL) morphology, in which lamellar and cylinder structures can be observed depending on viewing directions.²³ Therefore, if some grains with different orientations are adjacent to each other, both lamellar and cylinder structures are observed at the same time. However, STEM images are not clear enough to identify the morphology as HPL, probably because the microdomain structure was distorted by the crystallization of PVL block, as mentioned above. Although the detailed structural model of **ABC_{linear}** in the melt remains an open question, we emphasize again that the star-to-linear topology transformation drastically changes the microphase-separated structure. “

- The author's reply about the very low T_g (50 ° C) measured for PS is not satisfactory. The rather low molar mass of PS (8900 g/mol) is not sufficient to explain this value (see Flory-Fox equation for PS). In fact, by looking at Figure S21, the T_g of the PS block cannot be seen on the DSC graph. The data are noisy and there is no clear evidence of the C_p jump

characteristic of the T_g for PS. Hence the value cannot be determined. If the authors want to discuss T_g of this sample, another method to measure it should be used. The authors say that DMTA is not suitable for their sample, but I don't see why. Supported DMTA should work in any case. At the very least, shear rheology at different temperatures below and above the T_g should be provided.

As the reviewer pointed out, our DSC charts (Fig. S22) are rather noisy, but T_g can be confirmed from a shift of baseline, as indicated by green rectangles. However, we apologize for not sufficiently discussing the validity of T_g values. As the reviewer suggested, the very low T_g (50–60 °C) of PS blocks in linear **ABA** cannot be explained by molecular weight dependence of T_g of PS homopolymer. Therefore, PmVL block with much lower T_g should affect T_g of PS block. Judging from SAXS and TEM, linear **ABA** is speculated to form a less-ordered morphology with broad domain boundary. The decrease in T_g of linear **ABA** compared to star-shaped **A₂B₂** should be attributed to broadening of domain boundary by the topology transformation. Namely, a part of PmVL block segments near the block junction should penetrate the PS domain, resulting in the decrease in T_g of PS block (i.e., partial mixing in broad domain boundary).

Because our sample cannot be fixed on DMTA instruments, shear rheology as a function of temperatures was applied to our sample (Fig. S26). Both G' values in **ABA** and **A₂B₂** decreased with temperature. In the case of **ABA**, a slope of G' changed at 50 °C because of the glass transition of hard PS domains. In contrast, the specific T_g of **A₂B₂** was confirmed to be ca. 80 °C, around which a slope of G' clearly changed. Thus, DMTA results are consistent with DSC ones.

We have added data of shear rheology at different temperatures and comments in revised manuscript as follows; “We assume that the reasons for the less-ordered structure in linear **ABA** are (i) weaker phase segregation and (ii) weaker degree of block junction pinning at the domain interface, as compared to star **A₂B₂**. (i) For the triblock copolymer **A_nB_{2m}A_n** (n , m : degree of polymerization), the order–disorder transition temperature is much lower than that of the diblock copolymer **A_{2n}B_{2m}** (the same molecular weight as **A_nB_{2m}A_n**) and similar to that of **A_nB_m** (a half molecular weight of **A_nB_{2m}A_n**).²⁶ In addition, the phase segregation of star-shaped **(A_n)₂(B_m)₂** is stronger than that of **A_nB_m**.²⁷ Therefore, it is deduced from these facts that **A_nB_{2m}A_n** (**ABA** in this study) is in the weaker segregation state, as compared to star **(A_n)₂(B_m)₂** (**A₂B₂** in this study). (ii) Block junctions of star **A₂B₂** are expected to be more strongly pinned at the domain interface as compared to linear **AB**, because more severe thermodynamic and topological constraints should act on star **A₂B₂** than linear **AB**.²⁷ A similar interpretation would be valid for star **A₂B₂** and linear **ABA** (i.e., stronger block junction pinning of star **A₂B₂** than linear **ABA**), because two PS blocks (**A**) in linear **ABA** are apart

from each other by the intermediate PmVL block (**B**) and hence should be less correlated with each other. It is deduced from these two reasons that phase segregation becomes weak and domain boundary becomes broad by the topology transformation. Judging from SAXS (Figure 3) and TEM (Figure S24), linear **ABA** is speculated to form a less-ordered morphology with broad domain boundary. The decrease in T_g of linear **ABA** compared to star-shaped **A₂B₂** should be attributed to the broadening of domain boundary by the topology transformation (i.e., partial mixing of PmVL and PS blocks in broad domain boundary). ”

Reviewers' comments:

Reviewer #1 (Remarks to the Author):

The authors have satisfactorily answered to the raised issues. The manuscript may be published in its present state.